# Effect of Inappropriate Binder Grade Selection on Initiation of Asphalt Pavement Cracking

**Rodgers Mugume** [1,*] and **Denis Kakoto** [2]

[1]   Civil Engineering and Asset Management, Transport Research Laboratory, Wokingham,
      Berkshire RG40 3GA, UK

[2]   Technical Audit, Uganda National Roads Authority, Kampala P.O. Box 28487, Uganda;
      denis.kakoto@unra.go.ug

*   Correspondence: mugume_2rb@yahoo.co.uk; Tel.: +44-(0)79-0420-9112

**Abstract:** This paper is aimed at assessing in-service asphalt pavement cracking in order to fully understand its causes as well as reviewing the possible impact of implementing a new mix design method on failures observed. Field and laboratory investigations were conducted as well as a review of design and construction records. Substitution of a Performance Grade (PG) binder with straight run pen grade binder without performing independent Superpave PG verification tests greatly contributed to cracking. A 20/30 pen grade binder which was utilised had already undergone some change in its properties since its manufacture and therefore did not provide the required workability and crack resistance that it would otherwise have been expected to provide. Target mixing and compaction temperature ranges during construction were higher than those recommended confirming that the binder used had already undergone a change in its physical and chemical characteristics between the time of its manufacture and use. Additionally, a lapse in quality control and assurance during asphalt production and laying resulted in a highly voided wearing course which exacerbated the situation.

**Keywords:** cracking; mix design; superpave; asphalt concrete; bitumen; air voids; pavement performance

---

## 1. Introduction

In Low-Income Countries (LICs), recently there has been a deliberate prioritisation of investment in road infrastructure projects in a bid to enhance nationwide economic development and improve access to facilities and services. This is because 95% of cargo freight in the LICs is moved by road. However, one recurring challenge that has been encountered is deterioration of roads, especially asphalt-paved roads before the end of their service life—hence, premature pavement failure. Asphalt concrete is increasingly being preferred to surface dressings because of the increasing volume of heavily loaded vehicles being transported on the road network. Furthermore, recently there has been a drive in the road sector to shift from designing asphalt pavements using recipe or prescriptive specifications to performance-based design approaches owing to the perceived benefits associated with them. Historically, Marshall mix design method has predominantly been utilised in the design of asphalt mixes using familiar testing regimes and equipment. However, recently the Superpave Mix Design Method has been adopted on some few road projects in Uganda. That is to say Superpave Asphalt Binder Specification was employed for selection of the binder. One of those roads was the Rwentobo–Katuna road located in the South western part of the country at the border with Rwanda and Democratic Republic of Congo. It is part of the Mbarara–Ntungamo–Kabale–Katuna trunk road which is located in the tropical climate with monthly air temperatures ranging between 14.5–38 °C and an annual precipitation of 952 mm. The construction of trunk road was started in 2011 and final laying

of asphalt concrete was completed in 2016. It carries heavy traffic since it lies along the Northern Corridor road network linking the landlocked countries of the Great Lakes Region with the Kenyan maritime seaport of Mombasa. The road is surfaced with 150 mm of asphalt concrete comprising of 90- and 60-mm binder and wearing courses, respectively. The pavement layers include crushed stone base of 280–320 mm on top of an old pavement of about 200 mm mechanically modified lime-stabilised base. Pavement temperatures measured along the road at 20 mm below the pavement surface produced average maximum and minimum temperatures of 38 and 21 °C, respectively. However, the road has experienced significant defects in less than 5 years after completion including severe cracking and spalling of the road surface. The observed defects are often associated with roads nearing the end of their design service lives, thus in this case they have occurred prematurely.

This study was therefore focused on assessing the observed defects in order to fully understand their causes as well as reviewing the possible effect of implementing a new mix design method on the premature failures observed on the road. Field and laboratory investigations were conducted as well as a review of the design and construction records. The study has clearly highlighted the dangers of migration from recipe or prescriptive specifications to performance related specifications without an adaptation process first being put in place that suits a given country's local environmental and climatic conditions. Findings of the study will provide valuable lessons towards future design and production of asphalt mixes used for construction of asphalt pavements, in order to minimize or avoid similar problems observed on the road.

## 2. Literature Review

### 2.1. General

Pavement performance is a function of time and is usually linked to environmental and climatic conditions such as temperature and moisture as well as traffic loading [1]. Cracking is one of the significant in-service modes of deterioration for asphalt concrete pavements for which pavement engineers must consider during the design process of pavements [2,3]. It is greatly influenced by the quality of the bituminous mix which in turn is directly related to proper selection of materials including aggregates and bitumen. However, there are other factors that impact of the properties of bituminous mixes as well such as the temperature and duration at which the asphalt mixes are mixed, placed and compacted [4]. These factors are related to bitumen ageing which is normally determined by how much its physical and chemical characteristics change.

Bitumen quality is critical in producing a good mix since it coats and binds the aggregates together to produce a homogenous mix. Its properties are highly dependent on temperature because bitumen softens at high temperature resulting in permanent deformation while it stiffens and becomes brittle under low temperature resulting in cracking [4,5]. Bitumen therefore play a pivotal role in the deterioration of asphalt mixes and is directly responsible for most pavements defects such as rutting and fatigue cracking. In Uganda, bitumen has been extensively used in the road sector since it relatively cheap and provides good durability in asphalt mixes.

### 2.2. Mechanism of Ageing

Bitumen being an organic material undergoes ageing through the transformation in its mechanical properties, chemical composition and microstructure due to the impact of environmental conditions over its lifetime [6–9]. The transformation is mainly due to the oxidation process and loss of volatiles at high temperatures. The mixing stage of asphalt contributes the most rapid ageing phase and is significantly affected by the mix temperature, asphalt plant type and level of exposure of the bitumen to heat. It is widely known that asphalt production in the asphalt plant generally leads to a reduction in penetration of about 30 to 35%, implying that bitumen will usually harden by approximately one grade. Bitumen hardening is related to its mechanical properties and significantly impacts on the durability of asphalt mixes since it results in stiff and brittle mixes whose capacity to withstand stresses

and strains induced by traffic loading is significantly reduced. A change within the SARA (Saturates, Aromatics, Resins and Asphaltenes) fraction over time is known to impact on the chemical composition of bitumen [10–12]. Bitumen ageing is usually expressed in terms of short-term and long-term ageing. Rolling Thin Film Oven Test (RTFOT) is used to simulate short-term ageing while the Pressure Ageing Vessel (PAV) test is used to simulate long-term ageing. Short-term ageing is experienced within a few hours during the production, transportation and compaction of the asphalt mix, at relatively high temperatures of more than 130 °C often resulting in high oxidation rates. On the other hand, long-term ageing is a slow oxidation process that is usually experienced in the top few millimetres of the surface course in asphalt pavements in service due to exposure to traffic and climatic conditions during its service life [1]. UV radiation has been suggested as a contributor to long term ageing though its impact has been considered to be negligible [13,14].

### 2.3. Binder Selection

Selection of the appropriate binder is a very important aspect in the asphalt mix design process. Usually binder grades are affected by temperature and traffic loading expected in service, with stiffer binder grades selected when heavily loaded or slow-moving vehicles are expected. However, utilisation of stiffer binders often results in stiff mixes that have workability related issues since they are not easy to place and compact to the desired density. Furthermore, sometimes stiff binders are modified with polymers that further increase their viscosity hence substantially negatively impacting on workability at a given temperature [15,16]. Asphalt mixes with such stiff binders result in insufficient pavement compaction that often lead to significant performance-related defects resulting from high air voids. These include permeability of the pavement and age hardening of the binder due to oxidation hence significantly reducing the pavement life [17,18]. Since binder viscosity decreases with increasing temperatures, often the workability of such mixes is improved by applying higher heating temperatures. However, excessive increase in the heating temperatures to offset the stiffness of the mixes and therefore achieve the desired workability often has negative impacts on the binder such as hardening, damage of additives and increased volatile organic compounds. Research has shown that overheated bitumen oxides and hardens more as heating temperatures increase with Penetration Index values observed to decrease while at the same time viscosity values increase [19]. The heating process should therefore be controlled and monitored to ensure that it does not alter the basic and rheological properties of binders and asphalt mixes to an extent to cause the degradation of asphalt pavement durability [20,21]. This is usually achieved by using the binder viscosity to determine the appropriate mixing and compaction temperatures in order to avoid overheating of bitumen that would further contribute to its hardening hence significantly affecting the performance of the Hot Mix Asphalt (HMA) mixes [22]. However, with the increasing use of modifiers and new HMA mix types (e.g., Superpave and Stone Matrix Asphalt), problems with selecting satisfactory mixing and compaction temperature have been observed [23].

### 2.4. Effect of Air Voids

Asphalt pavement durability is also directly related to air voids in such a way that the density of a mix increases with decreasing percentage of air voids and vice versa. This is because the lower the air voids, the less permeable the mix becomes. A high percentage of voids results in a permeable mix with passageways through which air and water can enter the mix and damage the pavement while a low percentage can result in flushing. Voids Filled with Asphalt (VFA) is a property associated with air voids in a mix and basically is the percentage of voids in the compacted aggregate mass that are filled with asphalt cement. It is a measure of durability and has a significant correlation with the density of the mix in such a way that if VFA is too low, then there is not enough bitumen to provide the required durability [23]. HMA designed for moderate to heavy traffic may not pass the VFA requirement with a relatively low percent of air voids in the field even though the amount of air voids is within the acceptable range. Because low air void contents may be very critical in terms of resisting

permanent deformation, the VFA requirement helps to avoid those mixes that are susceptible to rutting in heavy traffic situations. VFA also restricts the allowable air void content for HMA that are close to the minimum VMA criteria. The dust to binder ratio is also an important property in the mix design process because it can greatly impact on the workability of asphalt mixes. Low dust-to-binder ratio values usually lead to tender mixes, which become stiff with increasing ratios. However, one must be careful not to have too high ratio since this is known to result in a tender mix as well which are susceptible to small stress cracking during compaction [24].

## 3. Materials and Methods

### 3.1. Selection of Study Sections

Study sections on which detailed investigations were undertaken, were selected by undertaking a drive through and walk-through reconnaissance of the roads. Drive through surveys were carried out at low speeds with frequent stopping to observe and assess the condition of failed and sound sections of road. Typical defects on the road were recorded and used to select sections that represented all the defects observed on failed sections. In order to obtain a better understanding of the nature and causes of the premature failures observed on the road, control sections experiencing similar conditions of traffic loading and road environment but exhibiting good performance were also selected for comparison purposes with failed sections.

### 3.2. Field Investigations

Site activities carried out on the selected sections of the road included roughness using the ROMDAS (Road Measurement Data Acquisition System) and visual condition surveys using a walked survey, rutting using straight edge [25], core sampling of the asphaltic surfacing using a core cutter, and trial pitting for sampling of asphaltic surfacing and pavement layers for laboratory testing. Furthermore, 100-mm-diameter cores were extracted mainly for strength tests (i.e., Marshall Stability and Indirect Tensile Strength) and 150-mm-diameter cores for observation of crack initiation, crack depth, visual quality assessments of the asphaltic layers, refusal density testing [26], and other material quality tests of the aggregates and the binder. The test pits measured 2 by 1 m in the transverse and longitudinal direction, respectively, were about 1.2 m deep from the top of the road surface across the outer wheel path which allowed for observation of the rutted layer(s). In situ density testing [27] of pavement layers was also carried in the test pits.

### 3.3. Laboratory Investigations

Material samples of asphaltic surfacing and pavement layers collected from the road were tested and analysed. Asphalt cores were tested for Indirect Tensile Strength (ITS) [28], Marshall Stability and flow [29]. Bitumen and aggregate recovered from the surfacing were tested to determine the properties of the bitumen and aggregate, and for comparison with specifications. Aggregate strength testing and determination of other quality characteristics such as particle size distribution was carried out for crushed stone or crusher run base and subbase [30,31]. Subgrade soil material was tested for plasticity index [32] and California Bearing Ratio (CBR) [33]. Gas-Liquid chromatography [34] was carried out on the recovered bitumen from the road as well as the representative bitumen samples from the UK for comparison.

### 3.4. Other Sources of Data

Axle load surveys were carried out along the road, analysed and then converted into Million Equivalent Standard Axles (MESA). Truck tyre pressures were also measured while conducting the axle load surveys. Traffic count as well as the design and construction data including reports were obtained from the road authority, and the information obtained was compared with measured data.

## 4. Results and Discussions

### 4.1. Field Investigations

#### 4.1.1. Detailed Sections

The main defects observed on Rwentobo–Katuna road were cracking of the asphaltic surfacing with varying extents and severities. Table 1 shows sections that were selected on the road after reconnaissance and a description of the main defects that influenced their selection for further detailed analysis.

**Table 1.** Detailed sections selected on the road.

| Section | Start Chainage | End Chainage | Length (m) | Remarks/Main Defects |
|---------|----------------|--------------|------------|----------------------|
| S1 | 96 + 200 | 96 + 400 | 200 | Severe crocodile cracking and spalling on both lanes |
| S2 | 97 + 750 | 98 + 050 | 300 | Severe crocodile cracking all over (but mostly on wheel paths) and patching. Some patches had cracks. |
| S3 | 100 + 150 | 100 + 450 | 300 | Several interconnected cracks in all lanes. Large width cracks on Katuna-bound climbing lane. |
| S4 | 125 + 950 | 126 + 000 | 50 | Control section with minimal cracks on Katuna-bound climbing lane and no cracks on other lanes |
| S5 | 156 + 050 | 156 + 200 | 150 | Flat control section with no cracks and appears to be overlay only |

#### 4.1.2. Roughness and Visual Condition Surveys

A roughness survey was conducted on the road in both directions and analysed using the International Roughness Index (IRI). Generally, a big portion of the road had IRI less than 6 which is indicative of a very good road condition. As shown in Figure 1, minor variations were observed between the left- and right-hand side probably because of inlay patches on the road.

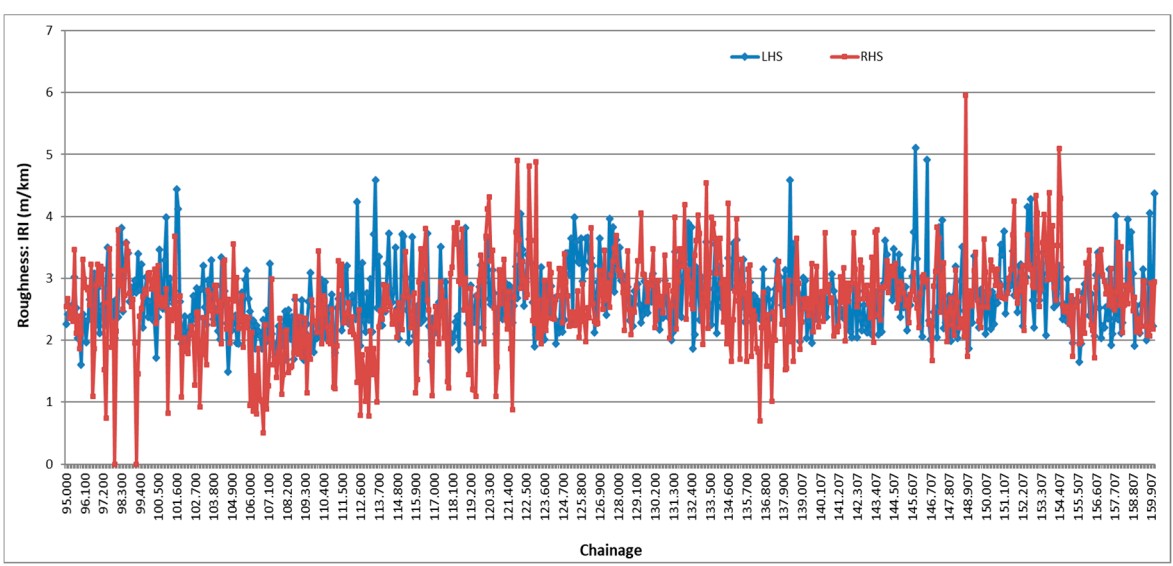

**Figure 1.** Roughness trend along the road.

Cracks on the road were evaluated based on their type, width, intensity, extent and position. As shown in Figure 2, cracking index (i.e., a product of crack intensity and extent) generally showed that the road had severe cracking especially in Sections 1 and 2.

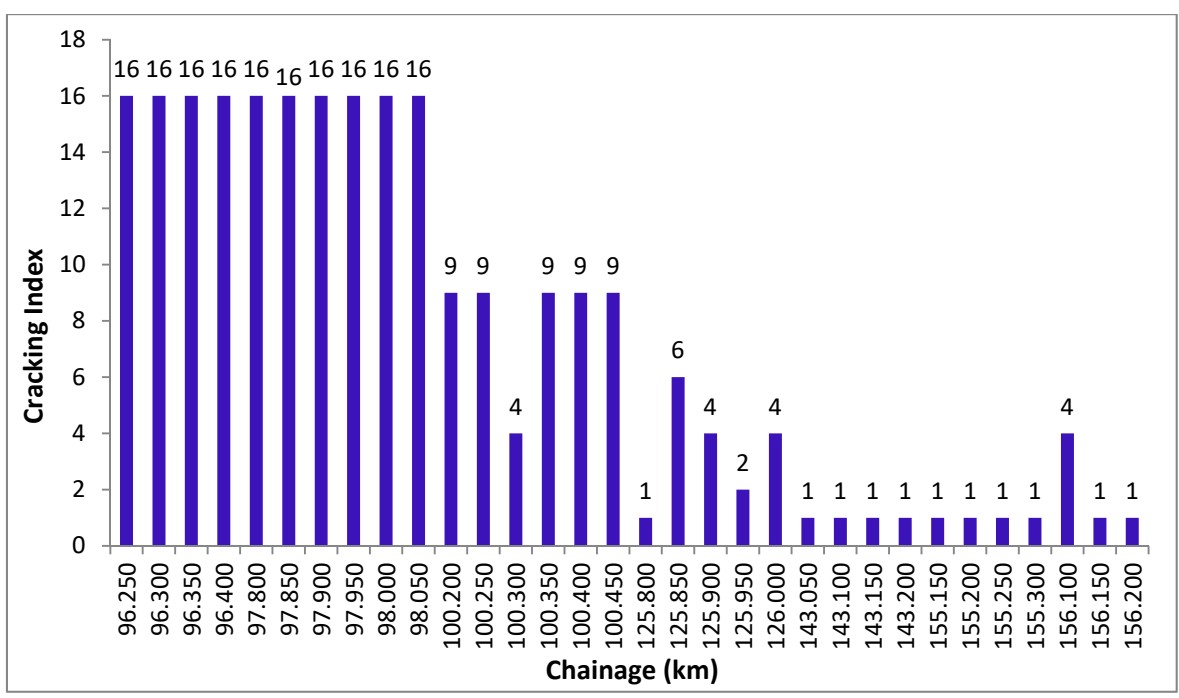

**Figure 2.** Cracking Index for the road.

### 4.1.3. Rutting

Rut measurements were carried out in both directions and processed in terms of mm of rut depth. As shown in Figure 3, highly variable rutting was experienced with occasional peaks indicating variability in performance either as a result of construction or traffic behaviour.

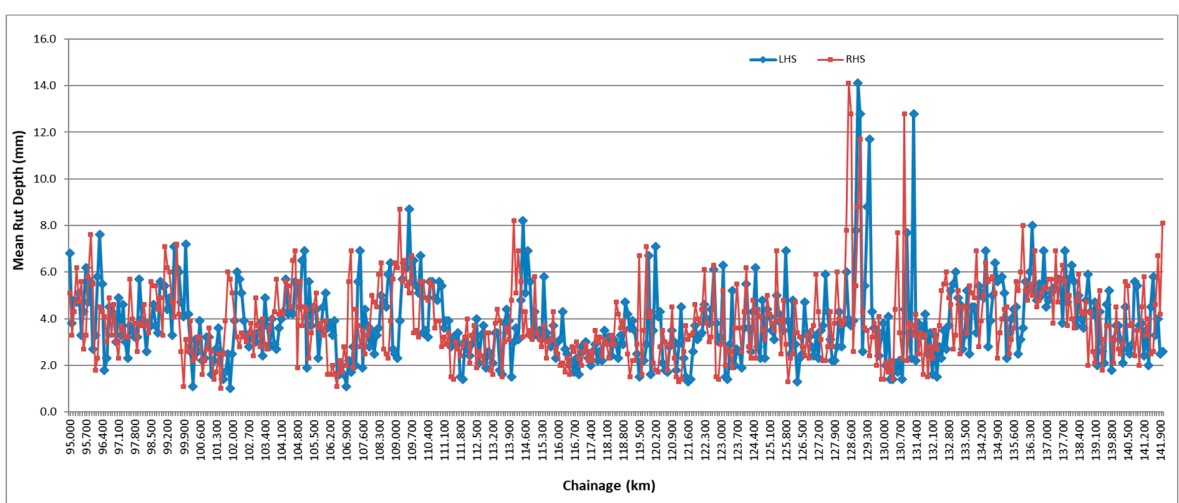

**Figure 3.** Rutting trend along the road.

### 4.1.4. Coring and Test Pits

Cracking Observation from Cores

Cores were extracted from the asphaltic surfacing before excavating test pits, with ones from cracked sections of the road being visually evaluated. As shown in Figure 4a, cracks were predominantly observed in cores obtained from the wheel path and were only confined to upper portions of the surfacing. It was further observed as shown in Figure 4b that the wearing course layer had de-bonded from the binder course. Construction records showed that the tack coat type used to bond the binder

and wearing courses was a stable 57% bitumen emulsion (grade CSS-1h) type and was applied at a rate of 0.55 L/m$^2$. It should, however, be noted that de-bonding was only experienced in cores obtained from the cracked sections of the road. Cores from sections with no or minimal cracking generally did not experience de-bonding hence ruling out the deficiency of the tack coat and cracking was also of lesser severity and intensity. However, as illustrated in Figure 5, cracking along longitudinal and transverse joints was observed and was widespread along the road. It must be emphasized that no cracking was observed in the binder course for all cores from both control and failed sections. This simply demonstrated that the cracking on road was confined to the surfacing layers and did not propagate throughout all layers of the pavement. A more detailed understanding of the causes of the cracking observed in the surface course is discussed in the results of laboratory testing section.

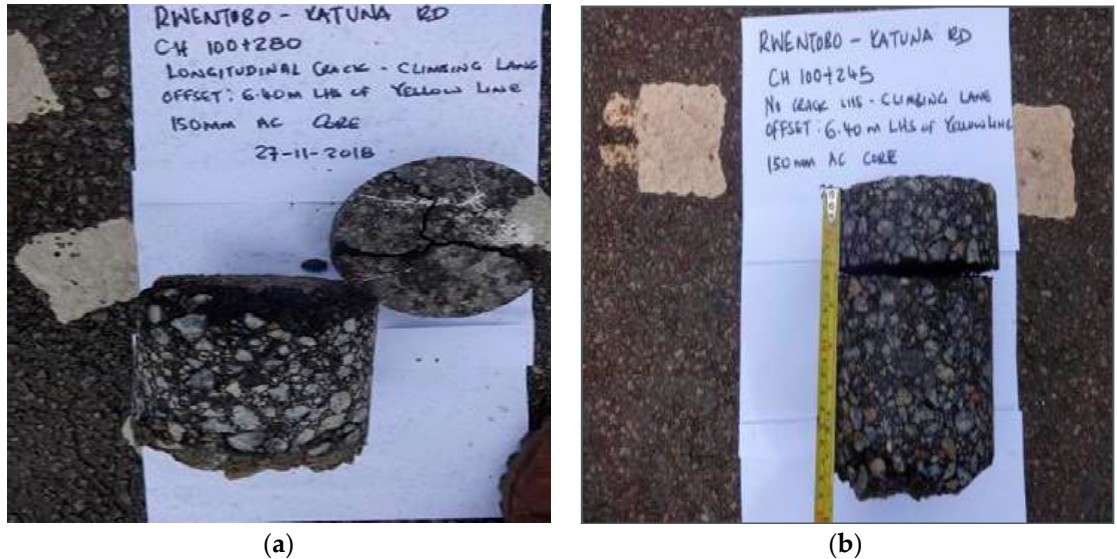

(**a**)                                                                 (**b**)

**Figure 4.** (**a**) Cracking confined to upper portions of the surfacing; (**b**) De-bonding of wearing course from the binder course.

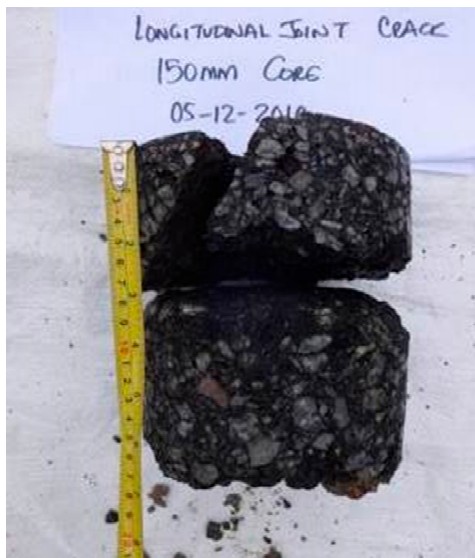

**Figure 5.** Dis-jointed longitudinal joint.

Test Pit Observations

Test pits excavated after removal of the asphaltic surfacing showed that the material composition for the base and sub-base layers was crushed stone and mechanically stabilised lateritic material,

respectively. Furthermore, minimal rutting was observed to be occurring on the road. As shown in Figure 6, test pits excavated clearly had a constant thickness of the pavement layers across the wheel path. This agreed with results from Falling Weight Deflectometer (FWD) tests carried out that showed the differences in deflection of the subgrade being negligible for the sections. The above finding indicated that rutting did not contribute significantly to the pavement deteriorations experienced on the road.

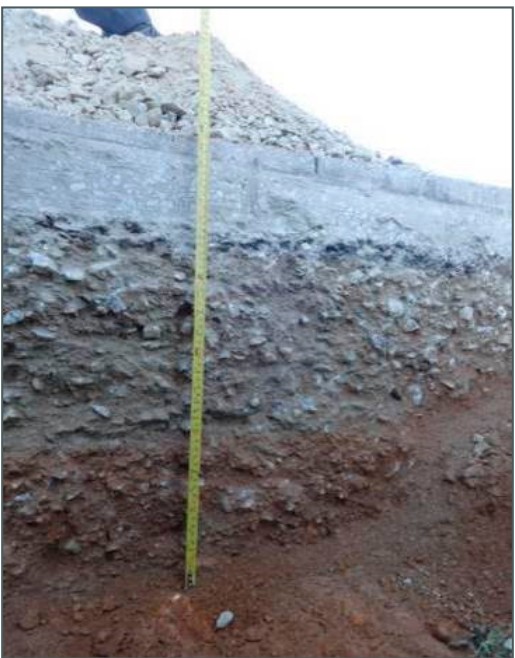

**Figure 6.** Test pit excavated in Section 1.

### *4.2. Pavement Design Analysis*

### 4.2.1. Traffic Assessment

Traffic assessment was carried out by comparing traffic data obtained for both the current counts and those carried out at design stage for trucks, semi-trailers and trailers as shown in Table 2. As expected, the actual growth rates determined using the current data differed from predicted values that were used at the design stage. This is because traffic prediction can be difficult and is usually not very accurate hence care should be taken to avoid gross under- or overestimation.

**Table 2.** Comparison of the design and actual growth rates.

| Vehicle Category | Number at Design Stage | Current Number | Actual Growth Rate (%) | Predicted Max. Growth Rate Used for Design (%) |
|---|---|---|---|---|
| Heavy Trucks three axle | 37 | N/A | 13.1 (est.) | 7.2 |
| Medium–Large Single Unit Trucks–Lorries | 37 | 127 | 13.1 | 7.2 |
| Truck trailers and semi-trailers | 37 | 133 | 9.0 | 7.2 |

Notes: est. = Estimated from next vehicle class, N/A = Current vehicle classification combines this with the next upper category.

### 4.2.2. Axle Load Surveys

Axle load surveys carried out along the road indicated differential loading by direction as shown in Table 3. It was observed that two-axle trucks generally had heavier axles than other truck categories with fifty percent of them having axles loads that were above the legal limit of 10,000 kg in both directions. Semi-trailers (five- and six-axle trucks) exhibit higher axle loads in the Katuna-bound direction. However, the 75th percentile load (the load exceeded by 25% of axles) in the Katuna-bound

direction is below the legal tandem axle load limit of 9000 kg/axle and only about 700 kg above the legal tridem axle load limit of 8000 kg/axle.

**Table 3.** Percentile axle loads.

| Trucks | Percentile | Towards Katuna | Towards Rwentobo |
|---|---|---|---|
| Two axle | 75th %ile (kg) | 11,800 | 11,300 |
| | 50th %ile (kg) | 11,400 | 11,100 |
| Three and four axle | 75th %ile (kg) | 8900 | 10,700 |
| | 50th %ile (kg) | 7600 | 8800 |
| Five and six axle | 75th %ile (kg) | 8700 | 8100 |
| | 50th %ile (kg) | 7900 | 6900 |

In order to compute traffic loading, Vehicle Equivalency Factors (VEF) were calculated from the axle loads using an equivalent single axle load of 80 kN as shown in Table 4. It was observed that the values used at the design stage were much higher than calculated values, implying that it is unlikely that overloading is to blame for the defects which are manifested on the roads. An estimated traffic loading of approximately 31 MESA was determined, which was lower than the actual design value of 72 MESA. This further indicated that the pavement structure that was designed should be able to adequately carry the current traffic loading.

**Table 4.** Comparison of design and current Vehicle Equivalence Factors (VEF).

| Vehicle Category | Designer's VEF | Measured VEF | Remarks |
|---|---|---|---|
| Medium Trucks (Two-axle Truck > 5 tonne) | 6 | 4.0 | |
| Heavy Truck (Three axles + four axles) | 19 | 7.7 | Currently very rare |
| Semi-Trailer | 13 | 6.2 | |
| Trailer | 25 | 7.6 | Currently very rare |

### 4.2.3. Tyre Pressures

Truck tyre pressures that were measured ranged between 576–1110 kPa with a median value of 900 kPa. These values are extremely high compared to standard tyre pressure ranges of 550–700 kPa that were assumed at design. It is widely known that tyre pressure significantly impacts on the fatigue performance of flexible pavements. Therefore, it is expected that such a combination of high tyre pressures and axle loads would result in significant damage to pavement surfacing. This further supports the phenomena of top-down cracking observed in the cores, since this type of cracking is usually associated with a combination of bitumen ageing and tensile stresses due to high tyre pressures.

### 4.3. Laboratory Investigations

### 4.3.1. Test Pit Materials

Materials obtained from the test pits were tested for moisture contents, densities and particle size distribution (PSD). Additionally, cores extracted from different sections of the road were tested for Indirect Tensile Strength (ITS) and Marshall characteristics of Stability and Flow. Test results from control and failed sections were all found to be very similar and within the acceptable specification requirements. This clearly showed that failures observed did not originate from the underlying layers below the asphalt surfacing. Furthermore, observations from PSD of aggregates recovered from the binder and wearing courses after extraction of bitumen showed that the recovered aggregates complied with the control points for Superpave nominal 12.5-mm mixes, as shown in Figure 7. This clearly indicated that aggregate gradation of the asphalt mix was not responsible for the observed deteriorations on the road. Therefore, the test results could not be used to identify potential failures or to isolate their causes on the road.

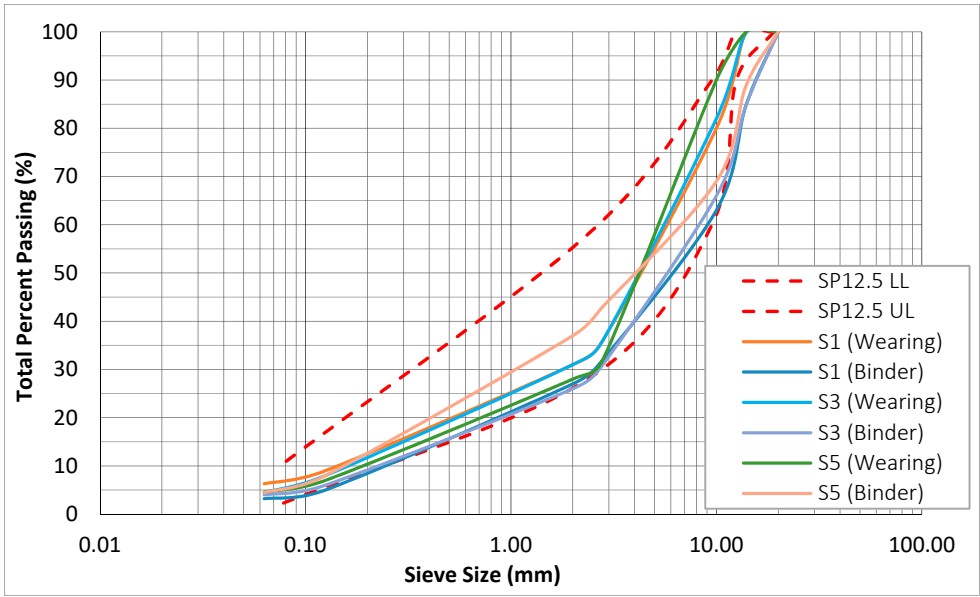

**Figure 7.** Particle size distribution of Binder and Wearing Course aggregates.

### 4.3.2. Volumetric Properties

Air Voids

Cored samples obtained from different sections of the road were subjected to refusal density compaction and their bulk specific gravities and theoretical maximum specific gravities were measured. The specific gravities obtained were used to determine the in situ air voids and air voids after Percentage Refusal Density (PRD). The specific gravities and bitumen content after extraction are shown in Table 5. It was observed that in situ air voids were generally high and, in most cases, higher than the recommended upper limit of 5%. Superpave mix design method recommends the design air voids specification range of 3–5%. Furthermore, it was noted that most of the air voids after refusal density compaction were close to the critical minimum of 3% and are likely to have contributed to the minimal rutting observed on the road. However, it was importantly noted that the asphalt mix laid was variable, as shown in Figure 8. Non-uniformity of compaction was measured in situ, whereby different points along the road had densities which were significantly higher or lower than the target value of 93.5% Maximum Dry Density (MDD). It is well-known that density has an inverse relationship with air voids and, therefore, low densities imply highly voided mixes, and vice versa.

It is well known that ageing of asphalt pavements in service is directly related to exposure to traffic and climatic conditions during its service life. This is more pronounced when the pavements have high in-place voids but is usually restricted to top few millimetres of the wearing course [1]. If initial compaction is adequate, then air voids will be reduced to acceptable levels, and as such the likelihood of bitumen hardening in the binder course will be minimized and not severe. It is therefore expected that points along the road with low densities had high voids and basically were the first to crack under stresses from temperature and traffic loading, and then later followed by those points with higher densities as a result of oxidisation and embrittlement of the bitumen. This supports the information from construction and maintenance records which showed that cracks manifested at different times through the Defects Liability Period (DLP) and beyond. It is understood that inappropriate mixes were corrected during construction and DLP and was manifested as patches on some sections along the road.

**Table 5.** Volumetric Analysis for Wearing Course.

| Section Description | Bulk Specific Gravity | Air Voids as Received | Air Voids after PRD | Bitumen Content |
|---|---|---|---|---|
| S1: Severe Crocodile Cracking and Spalling | 2.378 | 3.4% | 1.7% | 5.2% |
| S3: Severe Interconnected wide cracks | 2.284 | 7.9% | 2.7% | 6.7% |
| S5: Control with minimal/No Cracking | 2.287 | 7.9% | 3.2% | 4.4% |

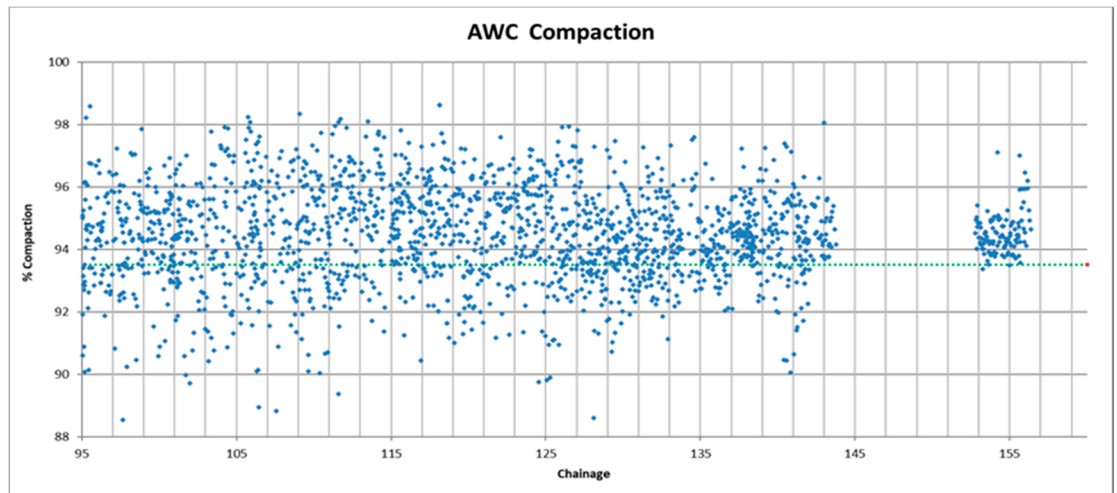

**Figure 8.** Compaction densities of the wearing course on the study road.

Voids Filled with Bitumen

Just like density, Voids Filled with Bitumen (VFB) are inversely proportional to air voids in a mix. VFB is also crucial in ensuring that asphalt pavements are not too voided and susceptible to damage by oxidation and water (rainfall), or too rich to cause bleeding and flushing. The target is just to have enough bitumen to adequately coat and hold aggregate particles together and prevent cracking but not too much bitumen to cause the pavement to bleed or to deform under traffic. Mix design methods including the Superpave method which was utilised for this road recommend an acceptable range of VFB values of 65–75%. For a country like Uganda with very high rainfall intensity, high temperatures and a high sun ultraviolet index, the mixes should be very close to the median of the range (i.e., 70%). However, as shown in Table 6, VFB results for all sections indicated that mixes were barely meeting the minimum value of 65. Moreover, some sections had high dust ratios compared with the acceptable range of 0.6–1.2. This is expected to result in stiff mixes of poor workability that are highly susceptible to cracking during compaction. It was also importantly observed that the minimum VMA requirement of 14% for nominal maximum size of 12.5 mm was achieved in almost all cases. This further emphasized the role played by VFB during mix designs in avoiding mixes that have marginally acceptable VMA values. It is expected that VFB influences the amount of asphalt content by limiting VMA values [24]. Therefore, though the mixes are passing the minimum VMA requirement, they are not adequate since they failed the VFB criteria.

**Table 6.** Wearing course VFB.

| Section Description | P0.075 | Pbe (%) | Dust ratio | VMA (%) | VFB (%) |
|---|---|---|---|---|---|
| S2: Severe crocodile cracking all over | 3.2 | 4.5 | 0.7 | 15 | 66 |
| (but mostly on wheel paths) | 4.8 | 3.7 | 1.3 | 13 | 65 |
| S5: Control with minimal/No Cracking | 4.0 | 4.4 | 1.1 | 16 | 51 |
| | 6.4 | 4.4 | 1.8 | 15 | 54 |

### 4.3.3. Bitumen Testing

Bitumen contents obtained after extraction from most cores were generally within the acceptable tolerance limits of the design bitumen contents of 4.3–4.9% for the wearing course as shown in Table 5. This indicated that the cracking phenomenon observed on the road was not directly related to the bitumen content. Though fatigue cracking is fundamentally related to inadequate pavement thickness, air voids and asphalt binder characteristics are also known to have a major impact on fatigue resistance of asphalt pavements. Therefore, in order to assess the likely impact of binder ageing on cracking observed, the recovered bitumen was tested for penetration and softening point and the results are shown in Table 7. Construction records indicated that a straight run 20/30 pen bitumen had been used on the project and had a penetration value of 26.6 dmm. Thus, for confirmatory purposes, results of penetration and softening point conducted on fresh (unaged) and RTFO-aged 20/30 paving grade bitumen manufactured by ORLEN Asfalt [35] are shown in Table 8. Furthermore, fresh (unaged) 15/25 penetration-grade bitumen samples were subjected to the Rolling Thin Film Oven Test (RTFOT) in order to simulate short term ageing on the binder and results are shown in Table 8. According to specifications [36], the retained penetration after RTFOT for 20/30 pen bitumen is 55%. This would imply that the retained penetration after RTFOT for the bitumen used on the project would equate to 14 dmm. Results of the 20/30 paving grade bitumen manufactured by ORLEN Asfalt as well as the confirmatory tests conducted on the 15/25 pen samples achieved the required minimum retained penetration by a wide margin as shown in Table 8. However, penetration results of the recovered bitumen from the road were well below 14 dmm as shown in Table 7. Penetration values were as low as 1 for the wearing course of the severely cracked sections, with the best performing control section (S4) having highest penetration value of only 7 dmm. This ideally would not be expected from the recovered bitumen since the road has been in service for less than 5 years and, therefore, the bitumen is not expected to have undergone such severe level of ageing. Furthermore, it was surprisingly observed that even the recovered bitumen from the binder course layers had unexpectedly undergone considerable hardening since it is expected that the wearing course would provide cover for the binder course.

**Table 7.** Penetration and Softening Point of recovered bitumen.

| Section Description | Penetration (dmm) | Softening Point (°C) | Layer |
|---|---|---|---|
| S1: Severe Crocodile Cracking and Spalling | 1.0<br>5.0 | 78.0<br>74.2 | Wearing Course<br>Binder Course |
| S3: Severe Interconnected wide Cracks | 4.0<br>11.0 | 78.6<br>67.0 | Wearing Course<br>Binder Course |
| S4: Control with very minimal cracks | 7.0<br>6.0 | 73.8<br>76.4 | Wearing Course<br>Binder Course |
| S5: Control with minimal/No Cracking | 2.0 | 79.8 | Wearing Course |

**Table 8.** Penetration and Softening Point before and after RTFOT.

| Fresh Sample Description | Tests | | | | | | |
|---|---|---|---|---|---|---|---|
| | | Penetration | | | Softening Point | | |
| Material | Brookfield Viscosity (Pa·s) | Before RTFOT (dmm) | After RTFOT (dmm) | Change in Pen | Before RTFOT (°C) | After RTFOT (°C) | Change in Softening Point |
| 20/30 paving grade bitumen from manufactured by ORLEN Asfalt | - | 27.9 [1] | 19.8 [2] | 8.1 | 62.1 [1] | 68.8 [2] | 6.7 |
| 15/25 penetration grade bitumen from UK | 1.734 | 16 | 12 | 4 | 69.8 | 75.6 | 5.8 |

[1] Average arithmetic means from tests on all production batches between 2011–2013; [2] Arithmetic means from tests conducted once per month in 2013.

### 4.3.4. Construction Records on Asphalt Production

Binder Selection

A Superpave Mix Design Method was adopted for the project implying that Superpave Asphalt Binder Specification was employed for selection of the binder. A PG 76–10 asphalt binder was recommended for use in the mix design because of its larger working temperature range, as well as the polymer's contribution towards asphalt resisting cracking in service. However, a straight run 20/30 pen binder was selected for use on the project instead of a PG 76–10 asphalt binder. This was justified based on the test certificates obtained from laboratories in the USA, which indicated that the 20/30 pen binder met the requirements of PG76–10. However, Superpave PG verification tests of the binder delivered to site were not carried out during construction since a full set of Superpave PG binder test equipment were not available. Only consistency tests that are not adequate to elaborate on the physical properties of the binder such as resistance to rutting and fatigue damage were done and used as a method of verification. Importantly, it is well known that Superpave as a design method derives its advantage from providing performance-based specifications for the asphalt and its constituents, particularly binders taking into account the local environment such as climate. It is for this very reason that the PG system was developed in the first case.

Additionally, binder ageing through oxidation process is one of the most critical characteristics related to binder behaviour. Oxidation is known to result in age hardening and embrittlement of the binder through the modification the structure and composition of the asphalt molecules. In the short term, oxidation rate increases considerably at higher temperatures due to loss of volatiles when the binder is exposed to elevated temperatures since it is known to be highly sensitive to temperature. This usually happens over a very short time within a few hours during asphalt mix production, transport and laying on site when the asphalt cement is heated to facilitate mixing and compaction [11,12]. In order to check the likely contribution of short-term ageing to cracking observed on the road, a review of the mixing and compaction temperature of the asphalt laid was undertaken as discussed below.

Mixing and Compaction Temperatures

Records from the bitumen manufacturer's certificate of conformity indicated that the recommended mixing and compaction temperature ranges of the 20/30 penetration grade bitumen used in the works were 165–175 °C and 110–140 °C, respectively. These temperatures corresponded to the mixing and compaction viscosities required for performance grade binders of 0.15–0.19 Pas and 0.25–0.31 Pas, respectively. However, the laboratory test results in the Mix Design Report indicated that the ideal mixing and compaction temperature ranges for the 20/30 binder were 178–183 °C and 172–175 °C, respectively.

Furthermore, construction records showed that target mixing and compaction temperature range for the works were 170–180 °C and 160–170 °C, respectively. These differed from the manufacturer's recommendation, especially the compaction temperature where the target range of 160–170 °C was used compared to the recommended range of 110–140 °C. This disparity raises a red flag and should have been reported to the bitumen manufacturer for further clarification and advice since it indicates that the bitumen had probably undergone some change since its manufacture. If mixing and compaction viscosities that were used at mix design to come up with the respective temperatures were similar to those used by the manufacturer, then this implies that the bitumen tested at the mix design stage had already undergone a change in its physical characteristics, probably through a loss of volatiles between the time of its manufacture and use. This was clearly manifested by a need for higher production or laying temperatures to achieve the required viscosities hence leading to further loss of volatiles in the process.

Additionally, Gas-Liquid chromatography was carried out on the recovered bitumen from the road as well as the representative 15/25 pen bitumen from the UK, as shown in Figures 9 and 10,

respectively. It was observed that the peaks chromatogram trace for the recovered bitumen showed a lower hump than that of the 15/25 pen bitumen, which indicated that the quantity of volatiles in the recovered bitumen sample was very low probably due to early ageing/oxidation.

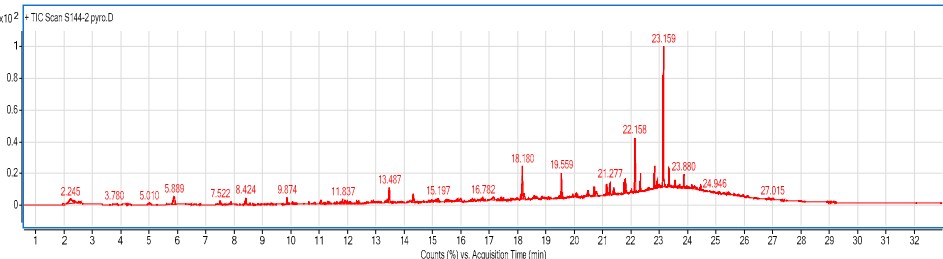

**Figure 9.** Chromatogram for bitumen recovered from the road.

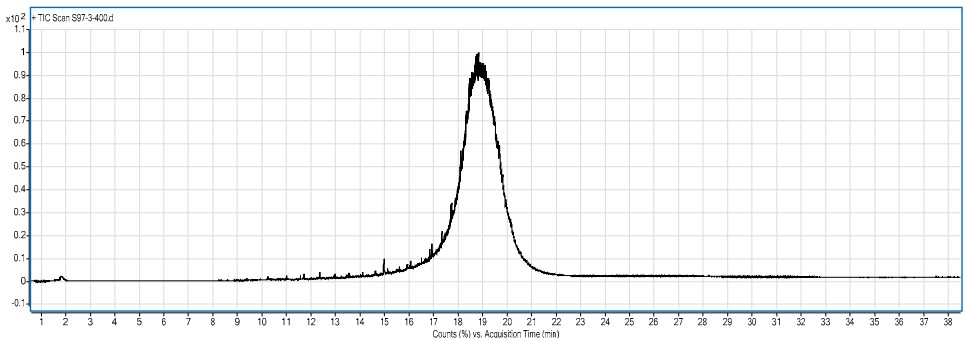

**Figure 10.** Chromatogram of 15/25 pen bitumen from the UK.

The Asphalt Institute Superpave Mix Design Manual (SP–2) recommends that the mixing and compaction temperatures should not exceed 165 °C and should not be lower than 115 °C, respectively [37]. Therefore, the target mixing temperature range used in the works of 170–180 °C did not comply with the specifications and is therefore expected to have contributed to the degradation of the bitumen. This is thought to be responsible for the severe cracking observed on the road since elevated temperatures are known to considerably increase the oxidation rate and embrittlement of the binder which usually manifests as cracks in asphalt pavements. It is therefore thought that the age hardening of the bitumen coupled with the heavy axle loading measured on the road resulted in wheel path cracks that progressively interconnected to ultimately became crocodile cracks as observed on the road.

Furthermore, construction records indicated that cracks were already manifesting on the vehicle lanes during the Defects Liability Period implying that the age hardening of the binder occurred during production and laying of the asphalt. This can only be attributed to overheating of the asphalt during production which usually occurs when a higher than recommended temperature is used or when a longer than necessary mixing duration is used during production. Though the method statement for production, laying, and compaction of the asphalt provided for a target production temperature of 170 °C, it did not have a provision for the batch mixing time. This was a serious oversight since it is well known that the mixing stage accounts for the most rapid ageing phase because bitumen is significantly impacted by the temperature of the mix, the type of asphalt plant and the level of exposure of the bitumen to heat. Mixing the asphalt for longer duration than specified, even at the correct mixing temperature easily leads to binder degradation. This is because during mixing, the bitumen film is thin and therefore easily degrades (embrittles) through oxidation. Moreover, the fact that other roads within the vicinity of this road did not exhibited such early crack initiation implied that the environmental conditions could not solely be responsible for the cracking observed.

Therefore, the above considerations generally pointed to a lapse in quality control and assurance especially in asphalt production and laying. The quality control mechanisms that were in place during construction were focused on the end result (laid asphalt) rather than the elements of the asphalt construction process such as appropriate binder selection, mixing and compacting temperatures, etc. This is expected to have played a critical role in the constructed asphalt layer not lasting for its intended life. Furthermore, there was a lack of capacity to implement the adopted Superpave Mix Design Method at the time of executing the project since some of the vital verification and quality control tests could not be carried out in-country due to lack of requisite equipment.

## 5. Conclusions and Recommendations

An investigation has been conducted to develop a better understanding of the defects observed on Rwentobo–Katuna road. Cracking was the major defect observed on the road in form of interconnected and extensive crocodile cracks that spread across the full width of the carriageway, as well as longitudinal cracks along the wheel paths. Cracking was confined to the wearing course layer of the surfacing with the other underlying layers performing well and their materials characteristics being within the acceptable specification requirements. Based on the analysis carried out, the following conclusions on the failures observed on the road can be drawn.

1. The substitution of PG 76–10 binder of known large working temperature range with straight run 20/30 pen grade binder without performing independent Superpave PG verification tests of the binder due to lack of a full set of Superpave PG binder test equipment was a grave oversight. Conducting only consistency tests on the 20/30 pen grade binder as a means of verification was inappropriate since these tests are not adequate and cannot elaborate on the physical properties of the binder like its resistance to rutting and fatigue damage. Superpave as a design method derives its advantage from providing performance-based specifications for the asphalt and its constituents, particularly binders taking into account the local environment such as climate. It is for this very reason that the PG system was developed in the first case.

2. The use of a 20/30 pen grade binder that had already undergone some change in its properties since its manufacture therefore did not provide the required workability and crack resistance that it would otherwise have been expected to provide. Furthermore, high dust ratios resulted in stiff mixes, hence compounding the poor workability. This made the asphalt more brittle and therefore more susceptible to cracking during compaction.

3. Target mixing and compaction temperature ranges for the works during construction were higher than those recommended by the manufacturer. This was especially for the compaction temperature whereby a target range of 160–170 °C was used compared to the recommended range of 110–140 °C. This confirmed that the bitumen used during construction had already undergone a change in its physical characteristics, probably through a loss of volatiles between the time of its manufacture and use.

4. Bitumen ageing/oxidation resulting in embrittlement of the binder during construction greatly contributed to cracking. This was confirmed by the very low penetration of the recovered bitumen. Bitumen ageing is thought to have been due to either overheating of the binder to temperatures well above the recommended mixing temperatures or heating at the required temperatures but for extended periods of time which is not in accordance with the specifications.

5. A highly voided wearing course with very low VFB values ranging between 51–66% as compared to the recommended mix design values of 65–75% is thought to have exacerbated oxidation of the binder. It is expected that the high voids led the bitumen in the asphalt to further oxidise/age rapidly in service and allowed water to easily penetrate the wearing course. Therefore, age hardening of the bitumen and penetration of water, coupled with the heavy axle loading and high tyre pressures measured on the road resulted in wheel path cracks that progressively interconnected to ultimately became crocodile cracks as observed on the road.

6. Future works on an adaptation process of performance related specifications to suit the country's local environmental and climatic conditions need to be undertaken to mitigate similar problems in the future.

Based on the findings of this investigation, the following recommendations are provided:

1. The use of Superpave Mix Design Method including application of Superpave Asphalt Binder Specification should be preceded by an adaptation process to suit a given country's local environmental and climatic conditions. Appropriate investment should be made to build capacity to implement Superpave Mix Design Method in prospective countries through training and procurement of appropriate equipment for conducting the vital verification and quality control tests. This will greatly contribute to migration from recipe or prescriptive specifications to performance related specifications that will provide linkage between quality characteristics (asphalt content, gradation, density, etc.), engineering properties (modulus, tensile strength, etc.), and performance (distress, serviceability level).
2. Since the laid AC was out of specification, quality control mechanisms should be put in place and enforced during construction that focus on the elements of the asphalt construction process such as appropriate binder selection, mixing and compacting temperatures rather than on only the end result (laid asphalt) in order to ensure that the constructed asphalt layer last for its intended life.
3. If a rehabilitation design is undertaken, a 60/70 pen bitumen is considered more appropriate utilising a modified Marshall Mix Design for the asphalt mix design since there is adequate capacity in terms of personnel and equipment to implement it. The rehabilitation design should take advantage of the strength of the existing underlying pavement layers which were found to be performing well to minimise the remedial costs.

**Author Contributions:** Conceptualization, R.M. and D.K.; Formal analysis, R.M.; Investigation, R.M. and D.K.; Methodology, R.M.; Project administration, R.M. and D.K.; Resources, R.M. and D.K.; Supervision, R.M. and D.K.; Validation, R.M.; Visualization, R.M.; Writing—original draft, R.M.; Writing—review and editing, R.M. and D.K. All authors have read and agreed to the published version of the manuscript.

**Funding:** This research received no external funding.

**Acknowledgments:** The authors would like to acknowledge Transport Research Laboratory (TRL) and Uganda National road Authority (UNRA) laboratories whose staff and facilities were used to conduct materials testing, and the guidance and briefings provided by A. Otto is greatly appreciated.

**Conflicts of Interest:** The authors declare no conflict of interest.

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
