# Peer review of "Effect of Inappropriate Binder Grade Selection on Initiation of Asphalt Pavement Cracking"

_sustainability, doi:10.3390/su12156099_

Round 1
Reviewer 1 Report
The paper is aimed at in-situ and laboratory investigating asphalt pavement cracking. The investigation has been conducted to develop a better understanding of the defects observed on the Rwentobo – Katuna road (Uganda). The work tackles an interesting topic and is suitable for the selected Special Issue “Pavement Design, Analysis and Material Characterization” as part of Sustainability Journal.
The work however needs the following improvements and clarifications before its approval. In the Reviewer's opinion, the following recommendations/clarifications should be considered:
- In section 1, before the State of the Art (SoA), the Authors should clarify what are the key novelties of this paper and the main contributions of this work beyond the current SoA. They are missing. This paragraph can be also included at the end of the literature review section.
- Main drawback of this work is its lack of scientific character. It looks like more a technical report than a research paper.
- More information are needed in the section Materials and Methods: this section is too short in the description of the analyzed materials and the adopted methods for investigating them. Please provide sufficient details to reproduce the experiments presented in this manuscript (to make it useful for other scientists). This information can be also placed in an appendix.
- Please add a last paragraph (to be added within the conclusion section) dealing with “future research steps”.
Author Response
Thank you very much for the comments and corrections provided. Below are our responses to the issues raised.
Point 1: In section 1, before the State of the Art (SoA), the Authors should clarify what are the key novelties of this paper and the main contributions of this work beyond the current SoA. They are missing. This paragraph can be also included at the end of the literature review section.
Response 1: The key novelty of the paper has been provided by a paragraph in the Introduction section as shown in Lines 59 – 62.
Point 2: Main drawback of this work is its lack of scientific character. It looks like more a technical report than a research paper.
Response 2: I understand your point of view that this paper may not be fully taking the shape of most traditional academic and theoretical papers. However, this paper is more focused on bringing out the practical issues and therefore tackle the real – world challenges of the engineering technology being encountered in the built environment. This is in line with the purpose of this Special Issue of the Sustainability Journal entitled “Pavement Analysis, Design, and Material Characterization” which is targeting the contribution of knowledge and experience from pavement engineers, material researchers, contractors, and users.
Point 3: More information are needed in the section Materials and Methods: this section is too short in the description of the analyzed materials and the adopted methods for investigating them. Please provide sufficient details to reproduce the experiments presented in this manuscript (to make it useful for other scientists). This information can be also placed in an appendix.
Response 3: The methods provided in the section of “Materials and Methods” are international standard methods and references have been provided as shown in Lines 159 – 179.
Point 4: Please add a last paragraph (to be added within the conclusion section) dealing with “future research steps”.
Response 4: Future research steps have been provided by a paragraph in the conclusion as shown in Lines 513 – 515.
Reviewer 2 Report
The manuscript covers an extensive field and laboratory experiments for the pavement with premature failures. However, without a field site constructed with PG 76-10 under the same climate and traffic condition, or perform a series of laboratory testing for PG 76-10 and 20/30 and compare their results, it is not reasonable to claim ‘the use of a 20/30 pen grade binder instead of a PG 76-10 did not provide for the workability and crack resistance to be realised’. There should be some data to support this conclusion in the manuscript. Unfortunately, this critical part is absent.
- In line 62, please check if to keep ‘due’ in ‘engineers must give due consideration during the design process’.
- In Figure 2, please mark the location of Sections 1 and 2.
- Please improve the quality of Figure 3: it’s kind of a blur.
Author Response
Thank you very much for the comments and corrections provided. Below are our responses to the issues raised.
It is true that it may not be reasonable for us to postulate that “the use of a 20/30 pen grade binder instead of a PG 76-10 did not provide for the workability and crack resistance to be realised”. Our interest was to point out that the 20/30 pen grade binder that had already undergone some change in its properties since its manufacture and therefore did not provide the required workability and crack resistance. The authors think that had the verification tests being conducted on the 20/30 pen grade binder, then this would have been detected. Revisions to reflect our agreement with your expert opinion are provided in the abstract (Lines 16 – 18) and in the conclusion (Lines 490 – 492).
Point 1: In line 62, please check if to keep ‘due’ in ‘engineers must give due consideration during the design process’.
Response 1: The sentence had been modified from “......pavement engineers must give due consideration during the design process...’ to “...pavement engineers must consider during the design process...” as shown in Lines 69 – 70.
Point 2: In Figure 2, please mark the location of Sections 1 and 2.
Response 2: Start and end chainages for all the sections have been added in Table 2 to ease their identification on the Figures 1 – 3. The authors thought that marking of the sections in figures would further congest them.
Point 3: Please improve the quality of Figure 3: it’s kind of a blur.
Response 3: The quality of Figure 3 has been improved by reducing the amount of data points.
Reviewer 3 Report
It is very nice to read an article that is focused on practical problem rather than theorotical issues because nowadays most scientific journals are starting to be mostly filled with papers that have very little connection with answering to real challenges of the engineering technology. Still, authors should adress to some deficiencies of their text or to add some comments:
- More details about the construction phase should be added, e.g. (a) when the paving works were completed, (b) how much time it took to finish the paving works, (c) in which part of the year the pavement layers were laid. We can only conclude now that significant defects occurred less than 5 years after completion (lines 47–48).
- What was the maximum grain size of the mineral mixture used for wearing course and binder course? In general, I would suggest to put separate paragraph with clear information about materials used during construction phase (if it is possible). Also, gradation curves of the designed mixtures would be very helpful.
- The authors should add more information about local climate, such as maximum and minimum temperature of the pavement that can occur in the discussed area. This knowledge is very important if Superpave system is being used or analyzed.
- Point "4.2.2. Axle load surveys" - what was the load of the standard axle? Was it 80 kN, 100 kN or maybe other value? It is hard to compare fatigue life of the described pavement structure without this information.
- The authors are not giving any information concerning interlayer bonding technique that was used during construction, or even if it was used at all. More details should be given what type of tack coat was used (bitumen emulsion? cut-back asphalt?), the amount of tack coat, etc., especially because as it is shown in figure 4 and 5, the wearing course was detached in many places from the binder course. Such behaviour of pavement layers may be a result of the deficiencies in the interlayer bonding.
- IIf such defects occured (alligator cracking) I would suggest at first to carry out evaluation of bearing capacity of the pavement by means of FWD or even Benkelmann beam. I know that according to the authors subgrade and subbase issues were not responsible for the pavement distress (as it is presented in point "4.1.4.2. Test pit observations"), but still it should be commented wheather such tests were carried out or not.
- Values presented in tables and statements in the text are sometimes contradictory or not commented enough by the authors. For example in table 5, bulk specific gravity, air void content and bitumen content for section 5 (classified as control section without or minimal cracking) are almost at the same levels as for section 3 (with severe interconnected cracks). It should be commented somehow. Also, according to the table 5, bitumen content for section 5 (control section) is much different than values of Pbe presented for the same section in table 6 (I am assuming, that "Pbe" means bitument content in table 6).
- In table 8 the authors are presenting for comparative purposes the results of tests of bitumen 15/25 subjected to RTFOT procedure, which was carried out in UK. I am concluding that this was done in this way because bitumen 20/30 was not avaible for testing. It would be better to compare values of penetration and softening point of recovered bitumen with those that can be found in literature, for example: https://www.orlen-asfalt.pl/PL/InformacjeTechniczne/PortalWiedzy/Documents/Poradnik_Eng_OSTATECZNY.pdf
- In my opinion, the main reason of the premature pavement distress was not wrong type of bitumen (20/30) but the fact that the bitumen was of general poor quality. Bitumen 20/30, produced in Poland can be classified as PG 82-16, and this working range is achieved without polymer modification (see the guidelines presented in the link above), which is well beyond limits specified in the design phase (PG 76-10).
Round 2
Reviewer 1 Report
The paper has been improved.
Reviewer 2 Report
I didn't see a change in the updated version compared with the original manuscript.
NA.
Reviewer 3 Report
Thank You for your extensive comments, answers and modifications of the paper, which now can be accepted in the present form.